# How Trait Gratitude Influences Adolescent Subjective Well-Being? Parallel–Serial Mediating Effects of Meaning in Life and Self-Control

**DOI:** 10.3390/bs13110902

**Published:** 2023-11-01

**Authors:** Yulin Li, Sige Liu, Dan Li, Huazhan Yin

**Affiliations:** 1School of Business Administration, Hunan University of Finance and Economics, Changsha 410205, China; liyulin@hufe.edu.cn; 2School of Education, Hunan Normal University, Changsha 410081, China; lsg15116263182@163.com

**Keywords:** trait gratitude, presence of meaning, search for meaning, self-control, subjective well-being, parallel–serial multiple mediation analysis

## Abstract

The relationship between trait gratitude and subjective well-being is well documented in the literature. Recently, growing attention has been given to examining which factors are determinants in the relationship. However, there are no studies to present a comprehensive model of how meaning in life and self-control jointly play a role in this relationship. This study investigated parallel and serial mediation of the presence of meaning, the search for meaning, and self-control in the relationship between trait gratitude and subjective well-being. A total of 764 adolescents (*M_age_* = 14.10, *SD* = 1.43, 48.43% males) from three middle schools in China completed a six-item measures of gratitude questionnaire form, a meaning in life questionnaire, a middle school students’ self-control ability questionnaire, and a satisfaction with life scale. The study revealed that trait gratitude affected the presence of meaning and subsequently affected subjective well-being. In addition, trait gratitude affected self-control through the presence of meaning and the search for meaning, and it subsequently affected subjective well-being. Therefore, the presence of meaning, the search for meaning, and self-control played an important role in the positive effects of trait gratitude on adolescent subjective well-being. The findings were in line with the intrinsic and extrinsic goal theory of gratitude and provided new insight to inform the improvement of adolescent subjective well-being in the future.

## 1. Introduction

Subjective well-being (SWB) is a crucial predictor of mental health, as it comprehensively assesses the self-perception of the quality of an individual’s life and reflects their social functioning and adaptation abilities [1]. Traditionally, SWB has referred to an individual’s overall assessment of their quality of life based on self-established standards, including two primary components: emotional and cognitive dimensions [2,3]. A higher level of SWB has been found to be a significant predictor of better physical health, stronger social relationships, and superior academic or work performance, while lower SWB was positively correlated with a higher incidence of problem behaviors, including Internet addiction, aggression, and emotional distress (i.e., depression, anxiety) [4,5,6].

In previous studies, the intrinsic and extrinsic goal theory explained the effect of gratitude on materialism and life satisfaction [7,8]. According to the theory of personality trait, trait gratitude is considered a positive personality trait that has a positive effect on subjective well-being [1,9]. As a crucial emotional trait of gratitude, trait gratitude refers to a life orientation that takes notice of the positive and appreciates it [10]. The intrinsic and extrinsic goal theory of gratitude also suggests that gratitude can enhance an individual’s SWB by promoting the satisfaction of their intrinsic needs [11]. Adolescents’ trait gratitude has been found to have a positive correlation with SWB in cross-sectional studies [12,13,14]. Meanwhile, longitudinal studies have also supported the notion that trait gratitude serves as a predictor of SWB [15,16]. Moreover, intervention studies targeting trait gratitude have affirmed that enhancing an individual’s level of trait gratitude can lead to a significant improvement in their SWB [10,17]. Adolescence is marked by significant physical, cognitive, social, and affective transformations attributable to rapid growth and maturation [18]. Within this age cohort, the development of trait gratitude demonstrates a consistent amelioration with advancing years [12]. In contrast to their senior counterparts, students in their early academic years tend to exhibit lower levels of trait gratitude [19]. Hence, within the scope of this study, examining the relationship between trait gratitude and subjective well-being, and the latent mediation variables among lower grades (age range = 12–17 years; mean age = 14.10 ± 1.43 years), will furnish fresh insights into the happiness experienced during this phase of life. Accordingly, we hypothesize that:

**H1.** 
*Trait gratitude has a positive correlation with subjective well-being.*


How does trait gratitude influence subjective well-being? The internal and external goal theory of gratitude posits that gratitude has the potential to foster the pursuit of intrinsic aspirations while diminishing the inclination toward extrinsic or materialistic objectives [12]. That is, a disposition towards gratitude has the potential to stimulate individuals’ meaningful internal objectives such as finding purpose in life, and it motivates them to strive towards these goals, ultimately enhancing their well-being [12]. Meaning in life (MiL) encompasses the subjective understanding, recognition, or perception of the purpose and significance of one’s existence, as well as an awareness of one’s life’s mission, purpose, and primary objectives [20]. Gratitude, as a substantial influencer of MiL, facilitates the acquisition of this sense of purpose [21,22]. For example, individuals who routinely document moments of gratitude demonstrate higher levels of MiL [23]. Prior studies have additionally shown that MiL significantly impacts subjective well-being, surpassing mere correlation [24,25,26].

Nevertheless, MiL encompasses two dimensions: the presence of meaning, which pertains to whether individuals perceive their lives as significant in a cognitive sense, and the search for meaning, which is a motivational component related to the desire to comprehend the significance of one’s existence. In terms of the presence of meaning, trait gratitude displays a positive correlation [27]. A disposition of gratitude fosters an intensified sense of significance in life, deriving from its independence from external objects and reliance on perceived value and importance. Those who experience gratitude are more inclined to discern the value of tasks, invest greater effort towards intrinsic goals, and consequently experience a heightened sense of the presence of meaning. Seligman [28] explained that well-being emanates from a life infused with purpose and meaning. Individuals possessing higher levels of meaningfulness not only experience greater happiness but also exhibit reduced susceptibility to depression and anxiety. Scholars have consistently discerned a robust positive correlation between the presence of meaning and subjective well-being [29]. In essence, individuals who acknowledge the presence of meaning tend to experience heightened positivity, including greater well-being and increased life satisfaction. As for the search for meaning, trait gratitude demonstrates a positive correlation [27]. However, the relationship between the search for meaning and subjective well-being has yielded inconsistent findings, potentially attributable to cultural divergences. Chinese individuals are predominantly shaped by Confucian cultural thoughts, which underscore a sense of duty in the world and promote a transcendent spirit in the face of setbacks. In the process of pursuing the meaning of life, it is an active and enjoyable attitude. Therefore, Chinese people tend to have higher levels of positive emotions such as subjective well-being when seeking meaning [25]. According to meaning management theory, people acquire the art of living a fulfilling life through the exploration and creation of meaning in their lives [30]. To sum up, trait gratitude exerts a positive influence on MiL, which stands as a pivotal influencing factor of subjective well-being. Therefore, we assume that:

**H2.** 
*The meaning in life plays a mediating role between trait gratitude and SWB.*


Additionally, *the intrinsic and extrinsic goal theory of gratitude* posits that a grateful life orientation contributes to enhancing resistance to external temptation, improving the ability to manage and control one’s life, ultimately leading individuals to greater satisfaction with their lives [12,31]. Self-control refers to the ability to overcome impulses, habits, or automatic responses and regulate one’s thoughts, emotions, and behaviors in order to achieve goals or adapt to external environments [32]. The literature suggests that positive emotions can enhance self-control. For instance, Dickens and DeSteno [33] found that gratitude could enhance self-control rather than inhibit it. That was because being grateful could enhance the inclination to relinquish current interests and pursue long-term goals, which necessitates suppressing impulsive responses to short-term objectives. Meanwhile, individuals with greater self-control are more likely to be well-adjusted and satisfied with their lives, as well as to achieve greater success and better psychological regulation [23,34,35,36]. The broaden-and-build theory postulates that positive emotions not only broaden one’s experiential scope but also foster the accumulation of valuable resources [37]. Consequently, this positive feedback loop culminates in an elevated sense of well-being. The theory of perceived control also posits that individuals with a heightened sense of control experience greater autonomy and well-being, while the absence of control may precipitate psychological distress such as depression [38]. In conclusion, the cultivation of gratitude may enhance the ability to exercise self-control and thereby contribute to elevated levels of well-being. In this regard, we assume that:

**H3.** 
*Self-control plays a mediating role between trait gratitude and SWB.*


Individuals who perceive their lives as imbued with meaning and value are more inclined to cultivate a robust internal drive for regulating their external behaviors and performances. This inclination leads them to transcend mere instinctual and impulsive modes of living [39]. The cognitive-affective system theory of personality [40] posits that personality traits (such as self-control) are structured according to three categories of social-cognitive factors: beliefs about control, values and aspirations, and strategies and competencies. A lucid and steadfast sense of meaning in life molds the objectives individuals pursue and the methodologies they employ in achieving them. This, in turn, delineates the focal point of self-control and provides it with impetus [41,42]. This implies that a sense of meaning in life serves as a motivator for individuals to adeptly regulate their emotions and behavioral patterns, all the while significantly enhancing their self-control [43]. However, individuals lacking MiL tend to be more susceptible to impulsive instincts. Recent studies have also indicated that MiL has a positive impact on self-control [44,45]. Individuals who possess a sense of meaning in life are better equipped to regulate their own behavior and emotional patterns. As an important psychological mechanism, self-control ability can be strengthened through increased MiL. Life meaning is a comprehensive set of values that provides guidance and motivation for adolescents to exercise self-discipline. Consequently, a heightened sense of the presence of meaning is expected to be associated with enhanced self-control [44]. As for the search for meaning, scholars have observed that individuals endowed with a sense of purpose and clear direction are inclined to tackle tasks with a sense of gravity, aligning their actions with their overarching life objectives through the reinforcement of self-discipline [46,47]. In sum, individuals’ self-control is influenced by their meaning in life. Therefore, our last hypothesis is that:

**H4.** 
*Meaning in life and self-control mediate the relationship between trait gratitude and SWB.*


## 2. Materials and Methods

### 2.1. Participants

A total of 764 Chinese adolescent students participated in the study. After omitting participants with incomplete data (i.e., missing values), in the data analysis, 737 valid questionnaires were obtained, including 357 (48.43%) males and 380 females (51.56%). They ranged in age from 12 to 17 years, with a mean age of 14.10 (*SD* = 1.43) years.

The current investigation obtained ethical clearance from the Research Ethics Committee at the primary author’s institution and adhered rigorously to the principles delineated in the Declaration of Helsinki. Informed consent was obtained from school authorities, parents or legal guardians, and the participating students.

### 2.2. Instruments

*Gratitude Questionnaire Six-Item Form (GQ-6)*. The Chinese version [48] of the 6-item self-report measure evaluates one’s propensity to experience gratitude (e.g., “Throughout my life, I am grateful for so many things”) [49]. A 7-point Likert scale ranging from 1 (strongly disagree) to 7 (strongly agree) was used. Higher scores indicated a greater trait gratitude level. Confirmatory factor-analytic results showed that the one-factor model fitted the data acceptably: *χ*^2^/*df* = 3.85, SRMR = 0.02, TLI = 0.97, CFI = 0.99, RMSEA = 0.06. In the current study, Cronbach’s alpha coefficient was 0.651.

*Meaning in Life Questionnaire (MLQ)*. The Chinese version of the MLQ [50] was used to evaluate MiL. The MLQ had two dimensions: the *presence of meaning* (e.g., “There is a clear purpose to my life”) and the *search for meaning* (e.g., “I am seeking something that will make my life more meaningful”), each with five items (Steger et al., 2009). A 7-point Likert scale ranging from 1 (absolutely false) to 7 (absolutely true) was used. The two dimensions were combined to generate a comprehensive global score. Higher scores indicated greater MiL. Confirmatory factor-analytic results indicated that the two-factor model fitted the data acceptably: *χ*^2^/*df* = 4.53, SRMR = 0.04, TLI = 0.94, CFI = 0.96, RMSEA = 0.07. The Cronbach’s alpha coefficient of the *presence of meaning* and the *search for meaning* in the present study were 0.805, 0.825.

*Middle School Students’ Self-control Ability Questionnaire*. This 36-item self-report measure evaluates one’s self-control [51]. A 5-point Likert scale ranging from 1 (strongly disagree) to 5 (strongly agree) was used. The questionnaire consisted of three dimensions: *the control over emotion* (e.g., “I get nervous when I take exams”) with eleven items, *the control over behavior* (e.g., “I have cheated in exams more than once”) with fifteen items, and *the control over thinking process* (e.g., “I’m easily influenced by the outside world”) with ten items. The three dimensions were combined to generate a comprehensive global score. Higher scores indicated a greater self-control. Confirmatory factor-analytic results indicated that the three-factor model fitted the data acceptably: *χ*^2^/*df* = 2.47, SRMR = 0.04, TLI = 0.90, CFI = 0.91, RMSEA = 0.04. In this study, Cronbach’s alpha coefficient was 0.899.

*Satisfaction with Life Scale (SWLS)*. The SWLS was used to assess the cognitive domain of subjective well-being [52] and measure individuals’ evaluations on how contented they were with their lives [53]. The scale included five items (e.g., “My life roughly conforms to my ideal”). A 7-point Likert scale (strongly disagree to strongly agree) was used to rate the statements. Higher scores signified greater SWB. Confirmatory factor-analytic results indicated that the single-factor model fitted the data acceptably: *χ*^2^/*df* = 3.85, SRMR = 0.02, TLI = 0.97, CFI = 0.99, RMSEA = 0.06. In this study, Cronbach’s alpha coefficient was 0.846.

### 2.3. Data Analysis

In the present study, data were analyzed using SPSS 21.0. The correlational analysis was carried out to determine the interrelationships among the variables. Mediation analyses of the relationship between trait gratitude, SWB, *the presence of meaning, the search for meaning*, and self-control were performed using the PROCESS version 3.3 procedure (Model 80) [54] in SPSS 21.0. Indirect effects were considered significant if the confidence intervals did not contain a value of 0, and the indirect effects standard errors were estimated using the bootstrap method with 95% confidence intervals (with 5000 samples) [54].

## 3. Results

### 3.1. Common Method Biases

In this study, some items of the questionnaires were reversely scored. In reference to previous studies [55,56], possible method biases were controlled in the testing procedure. The Harman single-factor test was used to conduct a statistical analysis of common method biases. The results showed that there were thirteen factors with characteristic roots greater than 1. The cumulative variation explained by the first factor accounted for 19.21%, which was less than 40%. Therefore, it could be concluded that there were no serious common method biases.

### 3.2. Descriptive Statistics and Correlations

The descriptive data and correlational coefficients are shown in Table 1. Correlational analyses demonstrate that SWB was positively correlated with trait gratitude, the *presence of meaning, the search for meaning*, and self-control. Trait gratitude was positively correlated with the *presence of meaning, the search for meaning*, and self-control. Furthermore, self-control was strongly associated with the *presence of meaning* and the *search for meaning*.

### 3.3. Hypothesis Testing

To test the parallel–serial mediation model between trait gratitude and subjective well-being, the PROCESS macro model 80 was adopted. As shown in Table 2, in stage 1, we examined the overall impact of trait gratitude on subjective well-being (H1). Trait gratitude was directly positively predicted SWB (β = 0.284, *p* < 0.001). In stages 2, 3, and 5, we scrutinized the mediating roles of both the presence of meaning and the search for meaning in the relationship between trait gratitude and SWB (H2). Trait gratitude was used as a predictor of the presence of meaning and the search for meaning (β = 0.311, *p* < 0.001; β = 0.320, *p* < 0.001), but only the presence of meaning (β = 0.268, *p* < 0.001) positively predicted SWB. In stages 4 and 5, we probed into the mediating role of self-control in the association between trait gratitude and subjective well-being (H3). Self-control positively predicted SWB (β = 0.177, *p* < 0.001), but trait gratitude did not positively predict self-control. In stages 2, 3, 4, and 5, we explored the serial mediating roles of meaning in life and self-control in the relationship between trait gratitude and subjective well-being (H4). Trait gratitude (β = 0.162, *p* < 0. 001), the presence of meaning (β = 0.268, *p* < 0.001), and self-control (β = 0.177, *p* < 0.001) all positively predicted SWB. See Figure 1 for details.

Table 3 shows the indirect effects of the *presence of meaning, the search for meaning*, and self-control. As shown, the total indirect effects of trait gratitude on SWB were significant (β = 0.124, CI = [0.081, 0.172]), accounting for 43.32% of total effects. Furthermore, there are three significant indirect effects of the parallel–serial multiple mediation. Firstly, the *presence of meaning* mediated the association between trait gratitude and SWB (β = 0.083, CI = [0.050, 0.121]), accounting for 29.08% of the total effects. The results partly confirmed hypothesis 2. Secondly, the result showed that trait gratitude affected SWB via the *presence of meaning* and then self-control (i.e., serial mediating effect) (β = 0.020, CI = [0.010, 0.034]), explaining 7.05% of the total effects (i.e., serial mediating effect). Thirdly, trait gratitude influenced SWB through the *search for meaning* and then self-control (i.e., serial mediating effect) (β = 0.006, CI = [0.009, 0.013]), explaining 2.09% of the total effects. Hypothesis 4 was confirmed.

## 4. Discussion

The current research proposes a parallel–serial mediation model that accounts for both direct and indirect effects between gratitude and SWB. Firstly, our findings support hypothesis 1, that higher levels of trait gratitude are positively associated with greater subjective well-being, in line with the *intrinsic–extrinsic goal theory of gratitude*. Additionally, these results replicate previous research indicating that grateful adolescents are more likely to experience a high quality of life [13,14]. Moreover, it could be inferred from the *broaden-and-build theory* [37] that gratitude can expand individuals’ cognitive flexibility and foster positive and enduring social and psychological resources, thereby promoting their SWB.

Secondly, the findings partially support hypothesis 2, that trait gratitude is positively associated with adolescent SWB through the mediation of MiL, which verifies *the intrinsic and extrinsic goal theory* again. As predicted, expressing gratitude may lead individuals to recognize the significance of their lives and subsequently enhance their levels of life satisfaction, which is consistent with the findings proposed by previous studies [21,27]. Trait gratitude enhances the perception of meaning in life by promoting the savoring of positive experiences and by strengthening social bonds, thereby providing a sense of purpose and connection [21,57]. At the same time, the linkage of gratitude to optimism, resilience, and openness to experiences motivates the desire to explore new possibilities and discover new sources of meaning and purpose [58,59,60]. In essence, the appreciative mindset cultivated through gratitude reveals existing meaning in one’s circumstances, while also inspiring the search for meaning by orientating individuals towards potential and growth [61,62]. Through this dual pathway of enhancing present meaning and motivating the search for meaning, the character strength of gratitude positively influences both the presence of, and the search for, meaning in life. This phenomenon may be observed in both Eastern and Western cultural contexts. Studies conducted in both Japanese and American cohorts have discovered that higher levels of gratitude are associated with adaptive psychological traits and heightened well-being, while lower levels of gratitude are linked to negative psychological processes and compromised emotional well-being [63]. Additionally, a disposition towards gratitude increases the likelihood of adopting positive coping mechanisms and social support, thus ameliorating psychological well-being in both the United States [64] and China [65]. This implies that the robust psychological benefits engendered by a disposition of gratitude may be consistent across diverse cultures [63]. Meanwhile, we found that both the *presence of meaning* and the *search for meaning* were positively correlated with well-being indices among Chinese students, which is consistent with previous meta-analyses [25,29]. However, the mediation role between trait gratitude and SWB is played solely by the *presence of meaning*, possibly due to its cognitive nature in contrast to the motivational aspect of *searching for meaning* [20,52]. This indicates that adolescents who possess a disposition of gratitude may be more likely to experience a heightened sense of purpose and exhibit better psychological adaptation. This indicates that grateful adolescents who experience a higher *presence of meaning* are more likely to exhibit better psychological adaptation, suggesting that the impact of trait gratitude on adolescent life satisfaction is primarily mediated by cognitive mechanisms. The findings are consistent with Watkins’ [66] proposition that grateful individuals tend to identify and amplify positive aspects of events, as well as reinterpret the meanings embedded in these situations, thereby enhancing their self-understanding and sense of meaning. Moreover, for Chinese individuals, life itself, to a considerable extent, embodies meaning; they do not require a distinct objective or mission to derive significance from life [67]. Through wholeheartedly embracing all life experiences, the Chinese may transcend the confines of the self, establishing profound connections and resonance with the entire universe. Such an outlook might compensate for the seemingly limited presence of religious beliefs among the Chinese, providing a pathway to transcendence akin to what religious beliefs offer in other cultures [67]. Therefore, higher levels of gratitude could enhance students’ well-being by raising the extent to which they realize the meaning in their lives.

Thirdly, this study has confirmed hypothesis 4, that both MiL and self-control serve as serial mediators in the relationship between trait gratitude and SWB among adolescents, which is consistent with *the intrinsic and extrinsic goal theory of gratitude* [11]. The findings suggest that grateful individuals experience greater happiness by enhancing their sense of life purpose and self-regulation. The possible reason is that grateful individuals are more likely to perceive the meaning of life, abandon immediate interests, and pursue long-term interests. During the process, individuals must assess multiple options and determine the most effective approach to their self-regulation abilities [68]. Adolescents who exhibit higher levels of self-control tend to employ more effective strategies in pursuit of personal goals [69], resulting in greater well-being and life satisfaction [1]. Consistent in part with previous studies [70], this study further demonstrates that not only the *presence of meaning* but also the *search for meaning* can facilitate students’ self-control improvement. The cognitive component of MiL exerts a greater influence than the motivational component, indicating that adolescents with high levels of meaningfulness possess clearer inner goals that drive them to regulate their thoughts, emotions, and behaviors in order to appreciate the value of life. Thus, trait gratitude may be a distal factor in SWB, while existential meaning and especially self-control are important proximal factors in SWB.

Last but not least, we opt to discuss hypothesis 3 after hypothesis 4, as our findings do not offer direct support for hypothesis 3. In other words, self-control does not appear to mediate the relationship between trait gratitude and SWB. However, this outcome does not negate the role of self-control in the influence of trait gratitude on subjective well-being. We ascertain that, when combined with the results of hypothesis 4, the influence of trait gratitude on self-control is entirely mediated by the presence of meaning and the search for meaning. This outcome underscores that the impact of trait gratitude on self-control is not direct, but rather mediated through other intermediary factors. Moreover, self-control significantly and positively predicts SWB. Individuals with a strong sense of self-control are more likely to exhibit stable emotions and higher self-efficacy when faced with stressful life events [58]. According to the psychological resilience theory, individuals with high psychological resilience possess more flexible cognition and richer coping strategies [71]. These abilities help reduce negative emotions and enhance positive emotions, and they thus predict an individual’s SWB. Therefore, the results indicate that individuals with high trait gratitude tend to possess stronger self-control, and those who perceive greater control over their lives are more likely to experience higher levels of SWB.

Enhancing SWB among adolescents is crucial, as it may act as a protective factor against clinical distress. This study investigates the determinants of SWB through the lens of goal-setting and attainment. Adolescents who exhibit gratitude are more likely to perceive meaning and inner goals by expanding their cognitive horizons and regulating their thoughts and behaviors towards achieving these objectives, thereby experiencing a profound sense of happiness. This implies that parents and educators have the potential to enhance adolescents’ subjective well-being by adopting the following perspectives. First, gratitude interventions, such as keeping a gratitude journal or practicing “naikan” meditation, could be effective means for adolescents to elevate their level of SWB. Second, career-planning guidance is a viable option, as there is ample evidence to suggest that it can enhance one’s sense of purpose in life. By cultivating this sense of purpose, adolescents are better equipped to identify and pursue their goals. Finally, parents and teachers can impart self-management, self-monitoring, and self-regulation techniques and strategies to students.

In summary, we broaden the scope of gratitude inquiry, moving beyond transient emotional states to encompass enduring personality traits. Our study delves into trait gratitude and its correlation with SWB. We pinpoint ‘meaning in life’ and ‘self-control’ as pivotal mediators, illuminating how trait gratitude can foster subjective well-being. This research elevates our comprehension of gratitude by clarifying the mechanisms through which individual disparities in grateful disposition can yield enduring subjective well-being. In essence, our emphasis on trait gratitude, SWB, and the role of ‘meaning in life’ and ‘self-control’ as distinctive mediators represents a significant extension of prior research and constitutes a substantial empirical contribution to the field.

Some limitations should be considered. Firstly, due to its cross-sectional nature, this study cannot establish causal relationships between trait gratitude and SWB with absolute certainty [16]. Therefore, future research employing longitudinal designs and experimental methods is necessary to validate these findings. Secondly, this study relies on self-reported questionnaires, which may be subject to response bias from participants, including social desirability effects [72,73]. Therefore, future research should incorporate a more diverse range of data collection methods, such as behavioral observation, interviews, or evaluations from parents and teachers. Finally, it is imperative to note that the homogeneity of our sample population, composed predominantly of early adolescents, may impose limitations on the generalizability of our research findings. Consequently, future studies should aim to encompass larger and more diverse samples drawn from various age groups and regions.

## 5. Conclusions

The current study has posited that adolescents who possess high levels of trait gratitude are more likely to experience greater life satisfaction. *The presence of meaning* has been found to partially mediate the relationship between trait gratitude and SWB, indicating that grateful students may have a heightened sense of purpose in life and consequently experience higher levels of SWB. Additionally, the *presence of meaning, the search for meaning*, and self-control have been identified as parallel–serial mediators between trait gratitude and SWB. Grateful students are specifically motivated to utilize their self-control abilities in pursuit of their goals by searching for, and perceiving, the meaning of life, resulting in increased well-being and fulfillment. In summary, the current research has made a significant contribution by providing new insights into the literature. It demonstrates that the presence of meaning, the search for meaning, and self-control independently and cumulatively mediate the relationship between trait gratitude and SWB.

## Figures and Tables

**Figure 1 behavsci-13-00902-f001:**
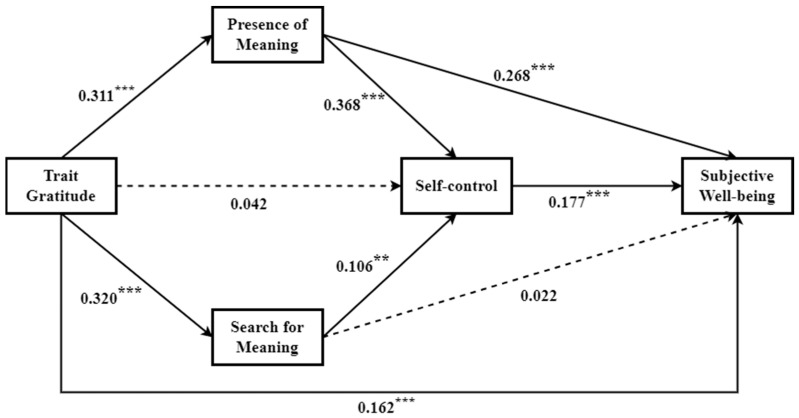
Parallel–serial multiple mediation model. ** *p* < 0.01, *** *p* < 0.001.

**Table 1 behavsci-13-00902-t001:** Descriptive statistics and correlations among variables.

Variables	*M*	*SD*	1	2	3	4	5
1. Trait gratitude	30.49	5.94	1				
2. Presence of meaning	20.20	6.42	0.331 ***	1			
3. Search for meaning	24.36	6.03	0.333 ***	0.348 ***	1		
4. Self-control	111.99	20.46	0.194 ***	0.424 ***	0.260 ***	1	
5. Subjective well-being	20.35	7.08	0.284 ***	0.402 ***	0.205 ***	0.332 ***	1

Note: *M* = mean; *SD* = standard deviation. *** *p* < 0.001.

**Table 2 behavsci-13-00902-t002:** Regression analysis.

Predictor Variables	Stage 1: SWB		Stage 2: Presence of Meaning		Stage 3: Search for Meaning		Stage 4: Self-Control		Stage 5: SWB	
	β	*t*	β	*t*	β	*t*	β	*t*	β	*t*
Trait gratitude	0.284	7.850 ***	0.311	8.164 ***	0.320	8.401 ***	0.042	1.066	0.162	4.162 ***
Presence of meaning							0.368	9.130 ***	0.268	6.210 ***
Search for meaning							0.106	2.626 **	0.022	0.551
Self-control									0.177	4.315 ***
*R* ^2^	0.080		0.100		0.105		0.189		0.212	
*F*	61.624 ***		66.653 ***		70.580 ***		46.290 ***		40.040 ***	

Note: ** *p* < 0.01, *** *p* < 0.001.

**Table 3 behavsci-13-00902-t003:** The indirect effects through the *presence of meaning, the search for meaning* and self-control.

	Effect	Boot SE	95% CI	Relative Mediating Effects
Total indirect effects	0.124	0.023	[0.081, 0.172] ^a^	43.32%
Trait gratitude → Presence of meaning → SWB	0.083	0.018	[0.050, 0.121] ^a^	29.08%
Trait gratitude → Search for meaning → SWB	0.007	0.015	[−0.023, 0.037]	
Trait gratitude → Self-control → SWB	0.007	0.008	[−0.007, 0.025]	
Trait gratitude → Presence of meaning → Self-control → SWB	0.020	0.006	[0.010, 0.034] ^a^	7.05%
Trait gratitude → Search for meaning → Self-control → SWB	0.006	0.003	[0.009, 0.013] ^a^	2.09%

Note: CI = bootstrapping confidence interval. ^a^ CI does not include zero.

## Data Availability

The datasets that support the findings of this study are available from the corresponding author upon reasonable request.

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
