# Peer review of "How Trait Gratitude Influences Adolescent Subjective Well-Being? Parallel–Serial Mediating Effects of Meaning in Life and Self-Control"

_behavsci, 2023, doi:10.3390/bs13110902_

Round 1
Reviewer 1 Report
Comments and Suggestions for Authors
Thank you so much for the opportunity to review this manuscript. The authors offer an interesting study on “How Trait Gratitude Influence Adolescent Subjective Well-being?A Parallel-serial mediating Model”.Overall, the idea of the research is quite interesting and the paper provide insightful results – however, there are some considerations I would like to see taken into account before publishing it.
1.In the introduction, it mainly elaborates on the relationship between various variables and the meaning of life and assumes the role of meaning of life in the model.The search for meaning has a more complex relationship with SWB or self-control compared to presence of meaning. The author should elaborate on the relationship between POM, SFM and SWB or self-control separately, rather than elaborate on it under the big concept of the meaning in life.
2. Who revised the Chinese version of MLQ did you use in the research? As far as I know, Gan et al.'s revised Chinese version only had 9 items, while Wang et al. revised a complete 10 items. Wang et al. also studied the structural validity and cross age measurement equivalence of the Chinese version of MLQ in the population aged 10-25.The author should provide specific information on the MLQ scale to ensure the clarity and scientificity of the measurement tool.
3. Why can trait gratitude positively predict students' presence of meaning and search for meaning respectively? Is this consistent or different between the Chinese and Western subjects? I think the author should enhance the analysis of the above problems.
4. Hypothesis 3 is not discussed.
Comments on the Quality of English Language-
Author Response
Reviewer 1:
Comments to the Author
Thank you so much for the opportunity to review this manuscript. The authors offer an interesting study on “How Trait Gratitude Influence Adolescent Subjective Well-being? A Parallel-serial mediating Model”. Overall, the idea of the research is quite interesting and the paper provide insightful results – however, there are some considerations I would like to see taken into account before publishing it.
In the introduction, it mainly elaborates on the relationship between various variables and the meaning of life and assumes the role of meaning of life in the model. The search for meaning has a more complex relationship with SWB or self-control compared to presence of meaning. The author should elaborate on the relationship between POM, SFM and SWB or self-control separately, rather than elaborate on it under the big concept of the meaning in life.
Response: Thank you very much for your suggestions. To present the potential relationships between the variables more clearly, we revised the manuscript and introduced its first from the overall framework of meaning in life, followed by elaborating on the relationship between POM, SFM and SWB or self-control separately.
“How does trait gratitude influence subjective well-being? The internal and external goal theory of gratitude posits that a disposition towards gratitude has the potential to stimulate individuals' meaningful internal objectives, such as finding purpose in life, and motivates them to strive towards these goals, ultimately enhancing their well-being (Bono & Froh, 2009). Meaning in life (MIL) encompasses the subjective understanding, recognition, or perception of the purpose and significance of one's existence, as well as an awareness of their life's mission, purpose, and primary objectives (Steger, 2009). Gratitude, as a substantial influencer of MIL, facilitates the acquisition of this sense of purpose (Kleiman et al., 2013; Liu et al., 2016). For example, individuals who routinely document moments of gratitude demonstrate higher levels of MIL (Tongeren, 2016). Prior studies have additionally shown that MIL significantly impacts subjective well-being, surpassing mere correlation (Doğan et al., 2012; Jin et al., 2016; Steger et al., 2009).
Nevertheless, MIL encompasses two dimensions: the presence of meaning, which pertains to whether individuals perceive their lives as significant in a cognitive sense, and the search for meaning, which is a motivational component related to the desire to comprehend the significance of one's existence. In terms of the presence of meaning, trait gratitude displays a positive correlation (Datu & Mateo, 2015). A disposition of gratitude fosters an intensified sense of significance in life, deriving from its independence from external objects and reliance on perceived value and importance. Those who experience gratitude are more inclined to discern the value of tasks, invest greater effort towards intrinsic goals, and consequently experience a heightened sense of the presence of meaning. Seligman (2002) explained that well-being emanates from a life infused with purpose and meaning. Individuals possessing higher levels of meaningfulness not only experience greater happiness but also exhibit reduced susceptibility to depression and anxiety. Scholars have consistently discerned a robust positive correlation between the presence of meaning and subjective well-being (Li et al., 2020). In essence, individuals who acknowledge the presence of meaning tend to experience heightened positivity, including greater well-being and increased life satisfaction. As for the search for meaning, trait gratitude demonstrates a positive correlation (Datu & Mateo, 2015). However, the relationship between the search for meaning and subjective well-being has yielded inconsistent findings, potentially attributed to cultural divergences. Influenced by Confucian culture, Chinese individuals tend to experience positive emotions when in pursuit of life's purpose (Jin et al., 2016). According to Meaning Management Theory, people acquire the art of living a fulfilling life through the exploration and creation of meaning in their lives (Wong, 2007). To sum up, trait gratitude exerts a positive influence on MIL, which stands as a pivotal influencing factor of subjective well-being.”
“Life meaning is a comprehensive set of values that provides guidance and motivation for adolescents to exercise self-discipline. Consequently, a heightened sense of the presence of meaning is expected to be associated with enhanced self-control (Li et al., 2019). As for the search for meaning, scholars have observed that individuals endowed with a sense of purpose and clear direction are inclined to tackle tasks with a sense of gravity, aligning their actions with their overarching life objectives through the reinforcement of self-discipline (Hedlund, 1977; Mackenzie & Baumeister, 2014).”
- Who revised the Chinese version of MLQ did you use in the research? As far as I know, Gan et al.'s revised Chinese version only had 9 items, while Wang et al. revised a complete 10 items. Wang et al. also studied the structural validity and cross age measurement equivalence of the Chinese version of MLQ in the population aged 10-25. The author should provide specific information on the MLQ scale to ensure the clarity and scientificity of the measurement tool.
Response: We apologize for the confusion here. Actually, we used the Chinese version of wang's (2013) MLQ, and we have added this information to the manuscript, thank you for your suggestion.
“The Chinese version of the MLQ (Wang, 2013) was used to evaluate MIL.”
- Why can trait gratitude positively predict students' presence of meaning and search for meaning respectively? Is this consistent or different between the Chinese and Western subjects? I think the author should enhance the analysis of the above problems.
Response: Thank you very much for your recommendations. We have revised the manuscript, incorporating a discussion in which we elaborate on Why can trait gratitude positively predict students' presence of meaning and search for meaning. Additionally, we explore the parallels in this relationship across Eastern and Western cultural contexts.
“Trait gratitude enhances the perception of meaning in life by promoting savoring of positive experiences and strengthening social bonds, thereby providing a sense of purpose and connection (Emmons & Stern, 2013; Kleiman et al., 2013). At the same time, gratitude's linkage to optimism, resilience and openness to experiences motivates the desire to explore new possibilities and discover new sources of meaning and purpose (Buddelmeyer & Powdthavee, 2016; Kumar et al., 2022; Szcześniak et al., 2020). In essence, the appreciative mindset cultivated through gratitude reveals existing meaning in one's circumstances, while also inspiring the search for meaning by orientating individuals towards potential and growth (Emmons & Mishra, 2011; Wong, 2011). Through this dual pathway of enhancing present meaning and motivating meaning-seeking, the character strength of gratitude positively influences both the presence of and search for meaning in life. In both Eastern and Western cultural contexts, this phenomenon may bear resemblances. Studies conducted in both Japanese and American cohorts have discovered that higher levels of gratitude are associated with adaptive psychological traits and heightened well-being, while lower levels of gratitude are linked to negative psychological processes and compromised emotional well-being (Srirangarajan et al., 2020). Additionally, a disposition to-wards gratitude increases the likelihood of adopting positive coping mechanisms and social support, thus ameliorating the psychological well-being in both the United States (Lin & Yeh, 2014) and China (Kong et al., 2015). This implies that the robust psychological benefits engendered by a disposition of gratitude may be consistent across diverse cultures (Srirangarajan et al., 2020).”
- Hypothesis 3 is not discussed.
Response: We apologize for this oversight. We systematically discuss the mediating role of self-control in Hypothesis 4. To present a clearer picture of the mediating role of self-control, we revised the manuscript and discussed Hypothesis 3 separately.
“Last but not least, we opt to discuss Hypothesis 3 subsequent to Hypothesis 4, as our findings do not offer direct support for Hypothesis 3. In other words, self-control does not appear to mediate the relationship between trait gratitude and SWB. However, this out-come does not negate the role of self-control in the influence of trait gratitude on subjective well-being. When combined with the results of Hypothesis 4, we ascertain that the influence of trait gratitude on self-control is entirely mediated by the presence of meaning and the search for meaning. This outcome underscores that the impact of trait gratitude on self-control is not direct, but rather mediated through other intermediary factors. Besides, self-control significantly and positively predicts SWB. Individuals with a strong sense of self-control are more likely to exhibit stable emotions and higher self-efficacy when faced with stressful life events (Buddelmeyer & Powdthavee, 2016). According to the psychological resilience theory, individuals with high psychological resilience possess more flexible cognition and richer coping strategies (Tugade et al., 2004). These abilities help reduce negative emotions, enhance positive emotions, and thus predict an individual's SWB. Therefore, the results indicate that individuals with high trait gratitude tend to possess stronger self-control, and those who perceive greater control over their lives are more likely to experience higher levels of SWB.”
Reviewer 2 Report
Comments and Suggestions for Authors
The authors used the questionnaire method to study the relationship between gratitude and subjective well-being, and the possible mechanisms involved. The author writes well in general, but some details should be supplemented and revised.
1. Some necessary references should be added. E.g., the definitions of SWB (Line 31-33) and trait gratitude (Line 41).
2. For participants part, the ethical body and whether the consent of the adolescent students' guardian has been obtained should be supplemented.
3. The Chinese version references of Gratitude Questionnaire, Meaning in Life Questionnaire the author used should be added.
4. For table 3, table cannot be spread across pages.
5. For Line 274, What’s ‘t relied on’?
Title: for “mediating”, “m” should be upper case.
Author Response
Comments to the Author
The authors used the questionnaire method to study the relationship between gratitude and subjective well-being, and the possible mechanisms involved. The author writes well in general, but some details should be supplemented and revised.
- Some necessary references should be added. E.g., the definitions of SWB (Line 31-33) and trait gratitude (Line 41).
Response: Thank you very much for your recommendations. We have added the necessary references to the manuscript.
“Traditionally, SWB has referred to an individual's overall assessment of their quality of life based on self-established standards, including two primary components: emotional and cognitive dimensions (Diener, 1994; Diener et al., 2010).”
“As a crucial emotional trait of gratitude, trait gratitude refers to “life orientation that take notice of the positive and appreciate it” (Wood et al., 2010).”
- For participants part, the ethical body and whether the consent of the adolescent students' guardian has been obtained should be supplemented.
Response: We apologize for the confusion here. Following your suggestion, we have added the relevant information to the manuscript.
“The current investigation obtained ethical clearance from the Research Ethics Committee at the primary author's institution and adhered rigorously to the principles delineated in the Declaration of Helsinki. Informed consent was obtained from school authorities, parents or legal guardians, and the participating students.”
- The Chinese version references of Gratitude Questionnaire, Meaning in Life Questionnaire the author used should be added.
Response: Thank you for your suggestion. We have added the Chinese version references of Gratitude Questionnaire, Meaning in Life Questionnaire to the manuscript.
“The Chinese version (Yu et al., 2011) of the 6-item self-report measure evaluates one's propensity to experience gratitude (e.g., Throughout my life, I am grateful for so many things) (McCullough et al., 2002).”
“The Chinese version of the MLQ (Wang, 2013) was used to evaluate MIL.”
- For table 3, table cannot be spread across pages.
Response: Thank you very much for your attentiveness, we have reformatted the manuscript to avoid spreading the table across the pages.
- For Line 274, What’s ‘t relied on’?Title: for “mediating”, “m” should be upper case.
Response: Your meticulous attention is greatly appreciated. 't relied on' should, in fact, read 'this study relied on.' We have rectified this oversight. Additionally, in line with your suggestion, we have capitalized the 'M' in 'Mediating' in the title.
Reviewer 3 Report
Comments and Suggestions for Authors
This manuscript (MS) explored the mechanism between trait gratitude and life satisfaction through a parallel-serial mediating model. The strengths of this MS include adopting a relatively large sample size and the effort to try to clarify the mechanism by using theories to guide the research.
My major concerns are also related to the theories.
1. This study's most used theory is the “internal and external goal theory of gratitude”. However, the authors did not introduce this theory clearly, leaving the scope and implications of the theories unclear. The authors should briefly introduce this theory, especially the “external” part which does not seem to be mentioned in the MS, then point out the related part to this study. Finally, the authors should cite whether other scholars try to use this theory to guide similar research like the current one. If there is, what’s this study’s unique contribution?
2. When the authors are trying to link trait gratitude to both the “presence of meaning” and “search for meaning”, and then to well-being, only links regarding the “presence of meaning” seem to be clear, whereas the links regarding “search for meaning” remain unjustified. I recommend that the authors reorganize these 2 paragraphs to make it clear (one for “presence of meaning” and the other for “search for meaning”).
3. The citation of the “Schachter-Singer two-factor theory of emotion” is inappropriate, as this theory has nothing to do with life meaning perception, and modes of living. The authors should correct this.
4. The authors also mentioned “broaden-and-build theory” (BBT) in the discussion part. I am curious why the authors do not use BBT to guide the variables’ relationship clarification. Self-control is considered a type of psychological resource as well.
5. Is trait gratitude the same thing as “expressing gratitude” and “exhibit gratitude”? The authors seem to use these terms interchangeably.
The authors should also strengthen the discussion part
1. What does “parents and schools must cultivate a positive learning environment” mean? It seems to have nothing to do with the implications mentioned above.
2. This study was conducted among a group of Chinese adolescents. Will the result be similar or different if in a different cultural group or age group (say, late adolescent)? The authors should address this population’s unique features.
Other minor issues
1. In the “Hypothesis testing” part, the authors mentioned that there are stages 1, 2,.., N. It’s unclear what was done in each stage.
2. The authors' usage of font size and type seem to be mixed sometimes, especially in lines 106-118, and 228-234.
Comments on the Quality of English Language
Many sentences read awkwardly, and the authors may consider asking a native speaker to proofread this MS before re-submission.
Author Response
Comments to the Author
This manuscript (MS) explored the mechanism between trait gratitude and life satisfaction through a parallel-serial mediating model. The strengths of this MS include adopting a relatively large sample size and the effort to try to clarify the mechanism by using theories to guide the research.
This study's most used theory is the “internal and external goal theory of gratitude”. However, the authors did not introduce this theory clearly, leaving the scope and implications of the theories unclear. The authors should briefly introduce this theory, especially the “external” part which does not seem to be mentioned in the MS, then point out the related part to this study. Finally, the authors should cite whether other scholars try to use this theory to guide similar research like the current one. If there is, what’s this study’s unique contribution?
Response: Thank you for pointing this out. We have added a brief description of the intrinsic and extrinsic goal theory, particularly elaborating on "extrinsic goals". According to this theoretical framework, gratitude has the potential to promote the pursuit of intrinsic aspirations while mitigating the propensity toward extrinsic or materialistic aims. We focus on the motivation of intrinsic goals. That is, a disposition towards gratitude has the potential to stimulate individuals' meaningful internal objectives. They exert more effort to achieve internal goals, while being less directed toward materialistic objectives, such as finding purpose in life, and motivates them to strive to-wards these goals, ultimately enhancing their well-being. Also, literature based on intrinsic and extrinsic goal theory is added to the text, and we point out the unique contribution of this study in the discussion section.
“In previous studies, the intrinsic and extrinsic goal theory explained the effect of grati-tude on materialism and life satisfaction (Lambert et al., 2009; Ma et al., 2017).”
“The internal and external goal theory of gratitude posits that gratitude has the potential to foster the pursuit of intrinsic aspirations while diminishing the inclination toward extrin-sic or materialistic objectives (Bono et al., 2022). That is, a disposition towards gratitude has the potential to stimulate individuals' meaningful internal objectives, such as finding purpose in life, and motivates them to strive towards these goals, ultimately enhancing their well-being (Bono et al., 2022).”.
“In summary, we broaden the scope of gratitude inquiry, moving beyond transient emotional states to encompass enduring personality traits. Our study delves into trait gratitude and its correlation with SWB. We pinpoint 'meaning in life' and 'self-control' as pivotal mediators, illuminating how trait gratitude can foster subjective well-being. This research elevates our comprehension of gratitude by clarifying the mechanisms through which individual disparities in grateful disposition can yield enduring subjective well-being. In essence, our emphasis on trait gratitude, SWB, and the role of 'meaning in life' and 'self-control' as distinctive mediators represents a significant extension of prior research and constitutes a substantial empirical contribution to the field.”
- When the authors are trying to link trait gratitude to both the “presence of meaning” and “search for meaning”, and then to well-being, only links regarding the “presence of meaning” seem to be clear, whereas the links regarding “search for meaning” remain unjustified. I recommend that the authors reorganize these 2 paragraphs to make it clear (one for “presence of meaning” and the other for “search for meaning”).
Response: I deeply appreciate the suggestions you have put forth. To present the potential relationships between the variables more clearly, we revised the manuscript and introduced its first from the overall framework of meaning in life, followed by introducing the mediating role of “presence of meaning” and “search for meaning” separately. Moreover, we have fortified the explication of the mediating role of “search for meaning” within it.
“How does trait gratitude influence subjective well-being? The internal and external goal theory of gratitude posits that gratitude has the potential to foster the pursuit of intrinsic aspirations while diminishing the inclination toward extrinsic or materialistic objectives (Bono et al., 2022). That is, a disposition towards gratitude has the potential to stimulate individuals' meaningful internal objectives, such as finding purpose in life, and motivates them to strive towards these goals, ultimately enhancing their well-being (Bono et al., 2022). Meaning in life (MIL) encompasses the subjective understanding, recognition, or perception of the purpose and significance of one's existence, as well as an awareness of their life's mission, purpose, and primary objectives (Steger, 2009). Gratitude, as a substantial influencer of MIL, facilitates the acquisition of this sense of purpose (Kleiman et al., 2013; Liu et al., 2016). For example, individuals who routinely document moments of gratitude demonstrate higher levels of MIL (Tongeren, 2016). Prior studies have additionally shown that MIL significantly impacts subjective well-being, surpassing mere correlation (Doğan et al., 2012; Jin et al., 2016; Steger et al., 2009).
Nevertheless, MIL encompasses two dimensions: the presence of meaning, which pertains to whether individuals perceive their lives as significant in a cognitive sense, and the search for meaning, which is a motivational component related to the desire to com-prehend the significance of one's existence. In terms of the presence of meaning, trait gratitude displays a positive correlation (Datu & Mateo, 2015). A disposition of gratitude fosters an intensified sense of significance in life, deriving from its independence from external objects and reliance on perceived value and importance. Those who experience gratitude are more inclined to discern the value of tasks, invest greater effort towards intrinsic goals, and consequently experience a heightened sense of the presence of meaning. Seligman (2002) explained that well-being emanates from a life infused with purpose and meaning. Individuals possessing higher levels of meaningfulness not only experience greater happiness but also exhibit reduced susceptibility to depression and anxiety. Scholars have consistently discerned a robust positive correlation between the presence of meaning and subjective well-being (Li et al., 2020). In essence, individuals who acknowledge the presence of meaning tend to experience heightened positivity, including greater well-being and increased life satisfaction. As for the search for meaning, trait gratitude demonstrates a positive correlation (Datu & Mateo, 2015). However, the relationship between the search for meaning and subjective well-being has yielded inconsistent findings, potentially attributed to cultural divergences. Chinese individuals are predominantly shaped by Confucian cultural thoughts, which underscore a sense of duty in the world and promote a transcendent spirit in the face of setbacks. In the process of pursuing the mean-ing of life, it is an active and enjoyable attitude. Therefore, Chinese people tend to have higher levels of positive emotions such as subjective well-being when seeking meaning (Jin et al., 2016). According to Meaning Management Theory, people acquire the art of living a fulfilling life through the exploration and creation of meaning in their lives (Wong, 2007). To sum up, trait gratitude exerts a positive influence on MIL, which stands as a pivotal influencing factor of subjective well-being.”
- The citation of the “Schachter-Singer two-factor theory of emotion” is inappropriate, as this theory has nothing to do with life meaning perception, and modes of living. The authors should correct this.
Response: Thank you for pointing this out, and to avoid any misunderstanding, we have removed the theory and revised the content of the manuscript.
“Individuals who perceive their lives as imbued with meaning and value are more inclined to cultivate a robust internal drive for regulating their external behaviors and performances. This inclination leads them to transcend mere instinctual and impulsive modes of living (Yek et al., 2017). The cognitive-affective system theory of personality (Mischel & Shoda, 1995) posits that personality traits (such as self-control) are structured according to three categories of social-cognitive factors: beliefs about control, values and aspirations, and strategies and competencies. A lucid and steadfast sense of meaning in life molds the objectives individuals pursue and the methodologies they employ in achieving them. This, in turn, delineates the focal point of self-control and provides it with impetus (Li et al., 2019; Tamir & Mauss, 2011). This implies that a sense of meaning in life serves as a motivator for individuals to adeptly regulate their emotions and behavioral patterns, all the while significantly enhancing their self-control (Schnell & Krampe, 2020).”
- The authors also mentioned “broaden-and-build theory” (BBT) in the discussion part. I am curious why the authors do not use BBT to guide the variables’ relationship clarification. Self-control is considered a type of psychological resource as well.
Response: Thanks to your suggestion, we cited the theory for the clarification of the relationship between variables in the mediating role of self-control section.
“The Broaden and Build theory postulates that positive emotions not only broaden one's experiential scope but also foster the accumulation of valuable resources (Fredrickson & Branigan, 2005)”
- Is trait gratitude the same thing as “expressing gratitude” and “exhibit gratitude”? The authors seem to use these terms interchangeably.
Response: Thank you for raising a good question. It is true that the terms "expressing gratitude" and "exhibiting gratitude" appear in some of the passages in the original text. There is a subtle difference between the three concepts. However, trait gratitude reflects a person's overall tendency or characteristic to express and exhibit gratitude. An individual who is high in trait gratitude will more naturally and frequently express gratitude through words and behaviors. Therefore, to a certain extent, trait gratitude can be regarded as a stable tendency to express and show gratitude.
- What does “parents and schools must cultivate a positive learning environment” mean? It seems to have nothing to do with the implications mentioned above.
Response: We agree with the reviewers. We have already made relevant suggestions from the perspective of gratitude, meaning in life, and self-control, and therefore, we have removed the above statement “parents and schools must cultivate a positive learning environment” from the manuscript.
- This study was conducted among a group of Chinese adolescents. Will the result be similar or different if in a different cultural group or age group (say, late adolescent)? The authors should address this population’s unique features.
Response: I appreciate your valuable input. The cohort under investigation in our study comprises students in their early academic years in the throes of adolescence. Compared to their senior counterparts, this demographic tends to exhibit lower levels of trait gratitude. However, this disparity provides a unique opportunity to delve into the relationship between trait gratitude and subjective well-being within this specific cohort. Admittedly, our study sample is circumscribed, and extending these considerations to other cultures and demographics necessitates further contemplation. We have taken steps to acknowledge these in “limitations” part; once again, thank you for your suggestion.
“Adolescence is marked by significant physical, cognitive, social, and affective transformations, attributed to rapid growth and maturation (National Academies of Sciences & Medicine, 2019). Within this age cohort, the development of trait gratitude demonstrates a consistent amelioration with advancing years (Bono et al., 2022). In contrast to their senior counterparts, students in their early academic years tend to exhibit lower levels of trait gratitude (Kashdan et al., 2009). Hence, within the scope of this study, examining the relationship between trait gratitude and subjective well-being among junior students (age range = 12–17 years, Mean age = 14.10±1.43) will furnish fresh insights into the happiness experienced during this phase of life.”
“Finally, it is imperative to note that the homogeneity of our sample population, composed predominantly of early adolescents, may impose limitations on the generalizability of our research findings. Consequently, future studies should aim to encompass larger and more diverse samples drawn from various age groups and regions.”
- In the “Hypothesis testing” part, the authors mentioned that there are stages 1, 2,.., N. It’s unclear what was done in each stage.
Response: We apologize for this confusion. Indeed, each stage serves to gauge the regression relationships among variables. Through the synthesis of test outcomes across these stages, we are poised to assess the validation of our hypotheses. In Stage 1, we examined the overall impact of trait gratitude on subjective well-being (H1). In Stages 2, 3, and 5, we scrutinized the mediating roles of both the presence of meaning and the search for meaning in the relationship between trait gratitude and subjective well-being (H2). In Stages 4 and 5, we probed into the mediating role of self-control in the association between trait gratitude and subjective well-being (H3). In Stages 2, 3, 4, and 5, we explored the serial mediating roles of the meaning in life and self-control in the relationship between trait gratitude and subjective well-being (H4). We have revised the manuscript to underscore the integrative functions of each stage. Once again, we extend our gratitude for your invaluable suggestions.
“To test the parallel-serial mediation model between trait gratitude and subjective well-being, PROCESS macro model 80 was adopted. As shown in Table 2, in stage 1, we examined the overall impact of trait gratitude on subjective well-being (H1). Trait grati-tude was directly positively predicted SWB (β = 0.284, p < 0.001). In stage 2, 3 and 5, we scrutinized the mediating roles of both the presence of meaning and the search for mean-ing in the relationship between trait gratitude and SWB (H2). Trait gratitude was used as a predictor of presence of meaning and search for meaning (β = 0.311, p < 0.001; β = 0.320, p < 0.001), but only the presence of meaning (β = 0.268, p < 0.001) positively predicted SWB. In stage 4 and 5, we probed into the mediating role of self-control in the association be-tween trait gratitude and subjective well-being (H3). Self-control positively predicted SWB (β = 0.177, p < 0.001), but trait gratitude not positively predicted self-control. In stage 2, 3, 4, and 5, we explored the serial mediating roles of the meaning in life and self-control in the relationship between trait gratitude and subjective well-being (H4). Trait gratitude (β = 0.162, p < 0. 001), presence of meaning (β = 0.268, p < 0.001) and self-control (β = 0.177, p < 0.001) all positively predicted SWB. See Figure. 1 for details.”
- The authors' usage of font size and type seem to be mixed sometimes, especially in lines 106-118, and 228-234.
Response: Thanks to the reviewer for the reminder. We have checked the journal requirements and have now harmonized the text according to the journal requirements.
- Many sentences read awkwardly, and the authors may consider asking a native speaker to proofread this MS before re-submission.
Response: Thank you for the reviewer's attention to the language fluency of the manuscript. We fully agree with the reviewer's suggestion. There are indeed some awkward sentences and language expressions in the paper. In the revision, we will invite native English-speaking colleagues to help us thoroughly proofread the entire manuscript to ensure language fluency and accuracy of expression. We will pay special attention to issues in grammar such as sentence structure, word choice, verb tense, articles, etc. At the same time, we will also invite peers with rich academic writing experience to help us revise sentences with stiff structures and ambiguities in expression from the content level. We will conduct comprehensive polishing and proofreading throughout the paper to ensure that the research viewpoints can be stated in standard, fluent and accurate English. Thank you again for the reviewer's comments. We will make efforts to revise accordingly and present the proofread English manuscript to you.
Reviewer 4 Report
Comments and Suggestions for Authors
This is a solid paper that makes a substantial contribution to the field. One may wonder about the novelty about researching for links between subjective well-being and gratitude. This need not, however, impede publication.
The authors could discuss how the Chinese situation may be different than other situations and expand on the Chinese cultural background.
Minor comments in attached pdf.

Comments on the Quality of English LanguageEnglish is wfine
Author Response
Comments to the Author
This is a solid paper that makes a substantial contribution to the field. One may wonder about the novelty about researching for links between subjective well-being and gratitude. This need not, however, impede publication. Minor comments in attached pdf.
The authors could discuss how the Chinese situation may be different than other situations and expand on the Chinese cultural background.
Response: We appreciate the reviewer encouraging us to consider and articulate the distinctiveness of the Chinese setting and how it informs interpretation of our results more deeply. We have expanded this part of the discussion in the revised manuscript.
“Trait gratitude enhances the perception of meaning in life by promoting savoring of positive experiences and strengthening social bonds, thereby providing a sense of purpose and connection (Emmons & Stern, 2013; Kleiman et al., 2013). At the same time, gratitude's linkage to optimism, resilience and openness to experiences motivates the desire to explore new possibilities and discover new sources of meaning and purpose (Buddelmeyer & Powdthavee, 2016; Kumar et al., 2022; Szcześniak et al., 2020). In essence, the appreciative mindset cultivated through gratitude reveals existing meaning in one's circumstances, while also inspiring the search for meaning by orientating individuals towards potential and growth (Emmons & Mishra, 2011; Wong, 2011). Through this dual pathway of enhancing present meaning and motivating meaning-seeking, the character strength of gratitude positively influences both the presence of and search for meaning in life. In both Eastern and Western cultural contexts, this phenomenon may bear resemblances. Studies conducted in both Japanese and American cohorts have discovered that higher levels of gratitude are associated with adaptive psychological traits and heightened well-being, while lower levels of gratitude are linked to negative psychological processes and compromised emotional well-being (Srirangarajan et al., 2020). Additionally, a disposition to-wards gratitude increases the likelihood of adopting positive coping mechanisms and social support, thus ameliorating the psychological well-being in both the United States (Lin & Yeh, 2014) and China (Kong et al., 2015). This implies that the robust psychological benefits engendered by a disposition of gratitude may be consistent across diverse cultures (Srirangarajan et al., 2020).”
“Besides, for Chinese individuals, life itself, to a considerable extent, embodies meaning; they do not necessitate a distinct objective or mission to derive significance (Zhang et al., 2016). Through wholeheartedly embracing all life experiences, the Chinese may transcend the confines of the self, establishing profound connections and resonance with the entire universe. Such an outlook might compensate for the seemingly limited presence of religious beliefs among the Chinese, providing a pathway to transcendence akin to what religious beliefs offer in other cul-tures (Zhang et al., 2016).”
- Original: Meanwhile, longitudinal studies have also demonstrated that trait gratitude served as a predictor of SWB (Unanue et al., 2019; Yang, 2021).
Comments: Too strong. rather 'supported'.
Response: Thank you very much for your recommendations. We have followed your prompting and replaced the similar expression with "Supported".
“Meanwhile, longitudinal studies have also supported that trait gratitude served as a predictor of SWB (Unanue et al., 2019; Yang, 2021).”.
- Original: Influenced by Confucian 79 culture, Chinese individuals tend to experience positive emotions when seeking life's purpose (Jin et al., 2016).
Comments: Why is this an influence of Confucian culture. Please elaborate.
Response: We apologize for the confusion here. We have revised the manuscript, elucidating the influence of Confucian philosophy in guiding the search for meaning and achieving subjective well-being among the Chinese populace.
“Chinese individuals are predominantly shaped by Confucian cultural thoughts, which underscore a sense of duty in the world and promote a transcendent spirit in the face of setbacks. In the process of pursuing the meaning of life, it is an active and enjoyable attitude. Therefore, Chinese people tend to have higher levels of positive emotions such as subjective well-being when seeking meaning”.
- Original: In the data analysis, 737 valid questionnaires were obtained, including 357 (48.43%) males and 380 females (51.56%).
Comments: Please specify what makes them invalid.
Response: Thank you for your suggestions. Due to partial incompleteness in the responses from some participants, we have excluded the data provided by these individuals. We have subsequently revised the manuscript and explicated the rationale for exclusion.
“A total of 764 Chinese adolescent students participated in the study. After omitting participants with incomplete data (i.e., missing values), in the data analysis, 737 valid questionnaires were obtained, including 357 (48.43%) males and 380 females (51.56%). They ranged in age from 12 to 17, with a mean age of 14.10(SD = 1.43).”.
- Original: 2.2. Instruments.
Comments: Please specify if all scales were validated.
Response: Thanks for the query raised by the reviewer. In accordance with the reviewer's counsel, within the revised manuscript, we have employed Confirmatory Factor Analysis (CFA) to substantiate the validity of the scales utilized, thereby ensuring the scientific rigor and applicability of the employed instruments. We extend our sincere gratitude to the reviewer for drawing our attention to this deficiency in the original manuscript.
“Gratitude Questionnaire Six-Item Form (GQ-6). The Chinese version (Yu et al., 2011) of the 6-item self-report measure evaluates one's propensity to experience gratitude (e.g., Throughout my life, I am grateful for so many things) (McCullough et al., 2002). A 7-point Likert scale ranging from 1 (Strongly Disagree) to 7 (Strongly Agree) was used. Higher scores indicated a greater trait gratitude level. Confirmatory factor-analytic results showed the one-factor model acceptably: χ2 /df = 3.85, SRMR = 0.02, TLI = 0.97, CFI = 0.99, RMSEA = 0.06. In the current study, Cronbach’s alpha coefficient was 0.651.
Meaning in Life Questionnaire (MLQ). The Chinese version of the MLQ (Wang, 2013) was used to evaluate MIL. The MLQ had two dimensions: the presence of meaning (e.g., “There is a clear purpose to my life”) and the search for meaning (e.g., “I am seeking some-thing that will make my life more meaningful”), each with five items (Steger et al., 2009). A 7-point Likert scale ranging from 1 (absolutely false) to 7 (absolutely true) was used. The two dimensions were combined to generate a comprehensive global score. Higher scores indicated greater MIL. Confirmatory factor-analytic results indicated that the two-factor model fits the data acceptably: χ2 /df = 4.53, SRMR = 0.04, TLI = 0.94, CFI = 0.96, RMSEA = 0.07. The Cronbach's alpha coefficient of the presence of meaning and the search for meaning in the present study were 0.805、0.825.
Middle School Students' Self-control Ability Questionnaire. This 36-items self-report measure evaluates one's self-control (Wang & Lu, 2004). A 5-point Likert scale ranging from 1 (strongly disagree) to 5 (strongly agree) was used. The questionnaire consisted of three dimensions: the control over emotion (e.g., “I get nervous when I take exams”) with eleven items, the control over behavior (e.g., “I have cheated in exams more than once”) with fifteen items, and the control over thinking process (e.g., “I'm easily influenced by the outside world”) with ten items. The three dimensions were combined to generate a comprehensive global score. Higher scores indicated a greater self-control. Confirmatory factor-analytic results indicated that the three-factor model fits the data acceptably: χ2 /df = 2.47, SRMR = 0.04, TLI = 0.90, CFI = 0.91, RMSEA = 0.04. In this study, Cronbach’s alpha coefficient was 0.899.
Satisfaction with Life Scale (SWLS). The SWLS was used to assess the cognitive do-main of subjective well-being (Diener et al., 1985) and measure individuals’ evaluations on how contented they were with their lives (Chen et al., 2022). The scale included five items (e.g., “My life roughly conforms to my ideal”). A 7-point Likert scale (strongly disagree to strongly agree) was used to rate the statements. Higher scores signified greater SWB. Confirmatory factor-analytic results indicated that the single-factor model fits the data acceptably: χ2 /df = 3.85, SRMR = 0.02, TLI = 0.97, CFI = 0.99, RMSEA = 0.06. In this study, Cronbach’s alpha coefficient was 0.846.”
Reviewer 5 Report
Comments and Suggestions for Authors
The detailed comments are in the attachment.

Comments on the Quality of English Languagenone
Author Response
Comments to the Author
1. “Please note that comments are attached to the document as postils”.
Response: We greatly appreciate the detailed and constructive comments you have provided. We have carefully read and addressed each of your suggestions point-by-point. For clear and efficient communication, we have responded directly within the comment boxes. We hope that our revisions meet your high standards. Thank you again for your time and effort in reviewing our manuscript. Your insights have helped improve the quality of our work.

Round 2
Reviewer 5 Report
Comments and Suggestions for Authors
Just a couple of minor suggestions to authors

Author Response
Thank you for your suggestions, and we have changed relative contents, such as "Mage" and "MiL".
Meanwhile, maybe we uploaded the wrong file by mistake for the suggestion of academic editor before, and now we have changed all in this file.
Thank you for your hard work.